# Multi-source heterogeneous blockchain data quality assessment model for enterprise business activities

**Haolin Zhang, Ran Zhang, Su Li, Likuan Du, Baoyan Song\*, Wanting Ji, Junlu Wang**

School of Information, Liaoning University, Shenyang, Liaoning, China

\* bysong@lnu.edu.cn

**Data Availability Statement:** The datasets are available from https://doi.org/10.6084/m9.figshare.25143503.v1.

## Abstract

Blockchain-based applications are becoming more and more widespread in business operations. In view of the shortcomings of existing enterprise blockchain evaluation methods, this paper proposes a multi-source heterogeneous blockchain data quality evaluation model for enterprise business activities, so as to achieve efficient evaluation of business activity information consistency, credibility and value. This paper proposes a multi-source heterogeneous blockchain data quality assessment method for enterprise business activities, aiming at the problems that most of the data in enterprise business activities come from different data sources, information representation is inconsistent, information ambiguity between the same block chain is serious, and it is difficult to evaluate the consistency, credibility and value of information. The method firstly proposes an entity information representation method based on the Representation learning for fusing entity category information (CEKGRL) model, which introduces the triad structure of related entities in blockchain, then associates them with enterprise business activity categories, and carries out similarity calculation through contextual information to achieve blockchain information consistency assessment. After that, a trustworthiness characterization method is proposed based on information sources, information comments, and information contents, to obtain the trustworthiness assessment of the business. Finally, based on the information trustworthiness characterization, a value assessment method is introduced to assess the total value of business activity information in the blockchain, and a blockchain quality assessment model is constructed. The experimental results show that the proposed model has great advantages over existing methods in assessing inter-block consistency, intra-block activity information trustworthiness and the value of blockchain.

## 1 Introduction

Blockchain [1] is a new distributed technology [2] for generating [3], storing, manipulating and verifying data through block-chain structure, consensus algorithm [4] and smart contract [5]. It can achieve value communication between trustless [6] nodes without relying on third-party trusted institutions [7]. These characteristics make blockchain widely used in enterprise

**Funding:** This study was supported by the General Program of University Basic Scientific Research of Education Department of Liaoning Province (Science and Engineering) (Initiating Flagship Service for Local Projects) (No. JYTMS20230761); the Applied Basic Research Program of Liaoning Province (No.2022JH2/101300250); the Ministry of Education University-Industry Collaborative Education Program (No. 230701160261310); the Nature Science Foundation Program Doctoral Startup Project of Liaoning Province (No. 2023-BS-085).

**Competing interests:** The authors have declared that no competing interests exist.

business activities, such as upstream and downstream supply chains, digital assets, business event monitoring, enterprise credit investigation, etc. Many IT companies around the world, such as IBM, Baidu and Alibaba, have established their own enterprise federated blockchain systems.

Blockchain is divided into public chain, private chain, and consortium chain based on its degree of decentralization. Public chain is completely open and transparent, not controlled by any organization, and everyone can participate; The write permission of private chains only belongs to individuals or a certain organization, with a high degree of centralization; The alliance chain, on the other hand, falls between the two and is only open to specific group organizations [8]. Participants can conduct transactions or access information, but only nodes in the alliance have the right to perform transaction verification, publish contracts, and other functions.

The quality of the information in existing enterprise blockchain systems varies. The information on business activities mostly originates from different fields and institutions, and the way of information representation is ambiguous. Moreover, it is difficult to ensure the credibility and value of the information in the block due to the constraints of the enterprise's own credibility and other conditions. Therefore, in the process of block establishment, there are problems of inconsistent information between blocks and inconsistent indicators such as credibility and value of content.

Moreover, traditional evaluation methods do not make full use of the features of blockchain. Blockchain features include leaving traces throughout the process, being non-temperable, and traceable. Evaluation efficiency and accuracy are low with traditional methods. This results in the inability of enterprise users and relevant regulatory authorities to quickly screen out suitable blockchains. Establishing a unified analysis model becomes challenging [9]. In the enterprise blockchain data layer, each distributed node encapsulates the business activity information received over a period of time into a data block and links it to the longest blockchain in the current blockchain network, forming a new block [10]. At the same time, with the increasing channels for obtaining information on enterprise business activities in blockchain, the amount of data is increasing. This information may originate from different institutions and fields, and there may be certain differences in the representation of information. Its credibility and information value cannot be measured, making it difficult to determine the overall quality of blockchain. Therefore, effective evaluation of enterprise blockchains is a hot spot and difficult area of research in the blockchain field.

To address these issues, this paper proposes a multi-source heterogeneous blockchain data quality assessment model for enterprise business activities to achieve an efficient assessment of the consistency, credibility and value of business activity information. The main contributions of this paper are as follows:

1. To address the problem of inconsistent and poor accuracy of enterprise business activity information in the blockchain, a semantic similarity calculation method between blocks based on contextual information is proposed to construct a triadic representation of enterprise activity information based on the CEKQRL model, introduce an attention mechanism [11] to realize the association between activity categories and activity information, increase the weight of key concern information, and construct a structural graph model through information context to improve the efficiency of similarity calculation, and then realize the information consistency assessment of the blockchain.

2. In response to the challenge of assessing the credibility of business information on the blockchain, we introduce a novel method for characterizing information trustworthiness that integrates the source, evaluative features, and content features. This method not only

verifies the credibility of the information source through an assessment of the trustworthiness of the publishing platform, page, and publisher but also generates a comment-based credibility representation by analyzing comment tendencies based on lexicon types [12]. Furthermore, it achieves a content-based credibility representation by examining the content characteristics of business activities. This holistic approach provides a comprehensive solution for information evaluation within the blockchain ecosystem.

3. To address the problem of value assessment of activity information in the blockchain, an information value assessment method is proposed to express the uncertainty of information through the magnitude of information, and then measure the amount of value of business operation activities in the block. Finally, a multivariate heterogeneous blockchain data quality assessment model for enterprise business activities is derived by integrating semantic similarity between blocks, blockchain content assessment and value assessment.

The rest of the paper is ordered as follows. A literature review section is provided in the Related Studies. In the Methods section, we introduce the calculation of the consistency, accuracy and value of blockchain information, and then conclude the quality assessment model. In the Experiments and Results section, we test the proposed model and discuss the experimental results. Finally, the Conclusion section is a summary of this paper.

## 2 Related studies

At present, many scholars have conducted in-depth research on data quality assessment methods and have achieved certain research results.

In blockchain information consistency evaluation, literature [13] proposed a structured gradient tree boosting (SGTB) algorithm for entity disambiguation method, which has better performance in cross-domain evaluation, but it ignores the role of contextual information of entities on the computation process; literature [14] proposed a factor graph-based inconsistent record pair computation method, which parses entities but does not correlate entity representation with the category it belongs to; literature [15] proposes a neural network with multi-view focus so as to capture more information features and improve disambiguation performance, but the method does not focus on entity contextual information and requires a large number of comparisons, which is less efficient.

In terms of blockchain information credibility assessment, literature [16] proposes a comprehensive credibility calculation method that effectively integrates multidimensional evaluation data, but focuses only on the relevant evaluation without paying attention to the information source as an influencing factor; literature [17] proposes a data credibility assessment method based on multi-source heterogeneous information fusion, which shows better results in improving the convergence of credibility calculation, but ignores the information content itself credibility; literature [18] proposes a supervised machine learning method for user-generated content credibility assessment, but the method only focuses on relevant comment information, ignoring the attention to information content and its sources.

In terms of blockchain information value assessment, literature [19] proposes a confidence-based reliability assessment method, which can accurately obtain the distribution to which the data belongs, but ignores where the value of the data itself lies; literature [20] combines variable weight theory and cloud model theory to construct a VW&ICM computational model for risk assessment, which weakens the influence of subjective factors on the assessment results, but the model focuses more on the weight assignment factors and ignores the information of the overall value of the data; literature [21] developed a new linguistic operation model to determine the overall weight and evaluation criteria from both subjective and objective

aspects, which overcomes the limitation of a single model but is only applicable to the case of large information uncertainty.

This paper introduces a novel approach: a multi-source, heterogeneous blockchain quality assessment model tailored for enterprise business activities. It addresses the limitations of current methods by prioritizing consistency, credibility, and data value [22]. This model enhances blockchain consistency assessment efficiency while also factoring in content credibility, yielding promising results even in scenarios with high information uncertainty.

## 3 Methods

As the information of business activities in blockchain mostly comes from different data sources, which leads to inconsistent information representation, such as data record format, entity and activity name designation methods. Moreover, the same enterprise entity will have different designations in different activities, and the same entity designation can refer to different entities in different contexts, resulting in the ambiguity of the business activity information stored in the blockchain and low data quality. To address this problem, this paper evaluates the consistency of blockchain data by comparing the semantic similarity between entities. The Fig 1 is the technical flowchart of this section.

### 3.1 Block entity representation based on CEKGRL model

The data related to enterprise entities are stored in the enterprise business activity blockchain, and in order to ensure the computational efficiency while effectively representing the information of this entity category, this paper proposes a block entity representation based on the CEKGRL model. Entity triad representation is introduced, and while learning triad knowledge, inconsistency detection of entity information is performed through accurate knowledge representation, and then potential correlations between entity categories and triad relationships are captured through attention mechanism, and entity similarity comparison is performed by combining the importance of different entity categories for a particular relationship and entity category information.

**3.1.1 Ternary information structure.**   The triple evaluation structure is a data model widely used for representing information, consisting of three components: subject, predicate,

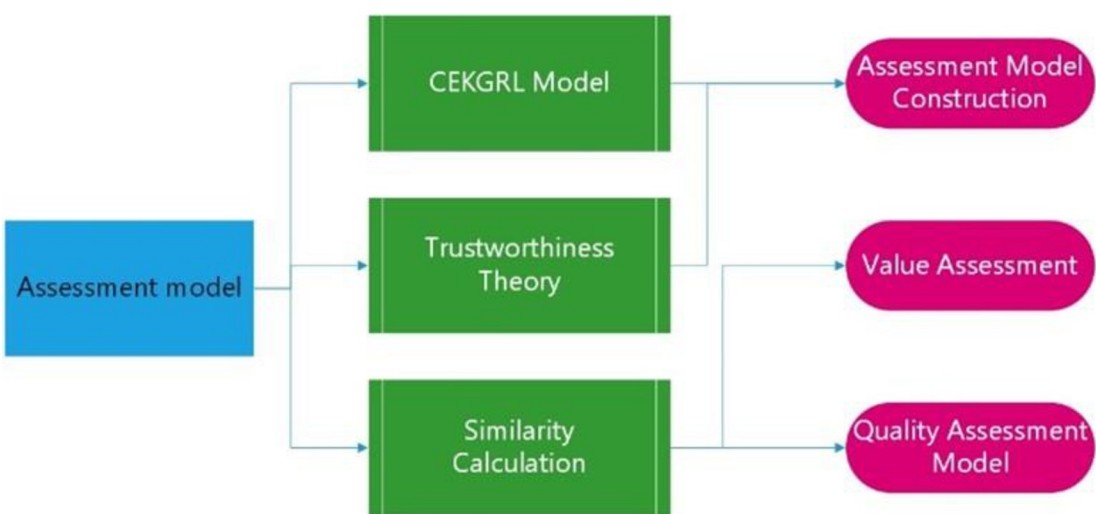

**Fig 1. The logic flowchart of showcase assessment model technical roadmap.**

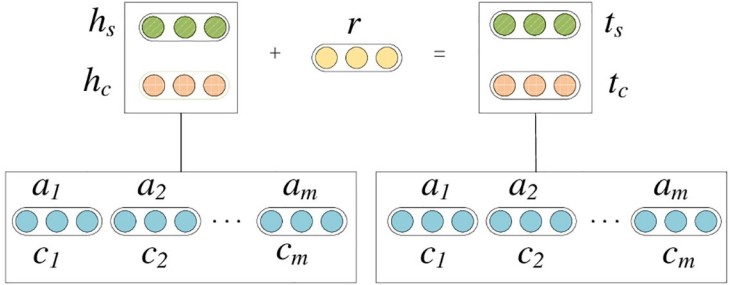

**Fig 2. Overall architecture of the CEKGRL model.**

and object. It is extensively applied in fields such as knowledge graphs and the semantic web. The triple evaluation structure is highly flexible and can be employed to represent various types of information. Different combinations of subjects, predicates, and objects can capture rich semantic relationships, making it suitable for diverse knowledge representation. In summary, the advantages of the triple evaluation structure lie in its flexibility, clear semantic representation, scalability, and effective expression of relationships, making it a key technology in the fields of knowledge representation and the semantic web.

Based on the CEKGRL model, business activity information is defined as $G = (E,R,S)$, where $E$ is the set of enterprise entities; $R$ is the set of enterprise relationships; $S \subseteq E \times R \times E$ denotes the set of triples, and the set of triples is represented by $(h,r,t)$, $h$, $r$ and $t$ represents the head entity (enterprise name), relationship (activity information direction) and tail entity (business activity information), respectively. In addition, this paper uses c to denote the activity categories and defines structure-based and category-based entity representations, representing the entity representation learned from the triad and the entity representation obtained by introducing the category representation, respectively. The overall architecture of the CEKGRL model is shown in Fig 2.

Among them, the ellipse composed of diagonal circles represents the structure-based vector representation, the ellipse composed of lattice-like circles represents the category-based vector representation, the ellipse composed of solid circles represents the vector representation of relationships, and the ellipse composed of hollow circles represents the vector representation of entity categories, a represents the attention fraction, and in order to integrate the two representation types, the energy function is defined in this paper as shown in Eq (1).

$$E = E_{ss} + \beta E_{cc} \tag{1}$$

$E_{ss} = \|h_s + r - t_s\|$ is the energy functions obtained by using structure-based entity representations for head and tail entities; $h_s$ and $t_s$ are the structure-based representations for head and tail entities, respectively; and the hyperparameters β is used to adjust the importance of the category-based representations in the CEKGRL model. $E_{cc} = \|h_c + r - t_c\|$ is the energy functions obtained by using category-based entity representations for head and tail entities. It should be noted that both the structure-based and category-based representations of entities use a unified relational representation r during the training process, ensuring that the vector representation space of both types can be unified by the same relational representation.

In the training process, the correlation between the entity category representation and the triadic relationship, i.e., the attention score, is first obtained through the attention mechanism, and then this attention score is used to weight and sum the category representation and use it as the category-based entity representation.

**3.1.2 Associated information categories and activity representation.** Different categories of enterprise entities can portray entities from multiple perspectives, and the same entity will focus on its different categories of information under different relationships, which is manifested by the different semantic relevance between different categories of the same entity and different relationships. In order to effectively utilize the potential correlation between the relationship and entity categories in the triad, this paper calculates and obtains the similarity between them by the following attention mechanism.

The essence of the attention mechanism lies in the ability to learn to ignore irrelevant information and focus on the key information when comparing similarities. In this paper, we first construct a scaled dot product attention model, combining the CEKGRL model with relationships $r$ as *query* vectors and categories $c$ as both vectors *key* and *value* vectors. In the implementation process, in order to speed up the processing efficiency, the attention is calculated in the form of a matrix, and the representation vectors of multiple inter-enterprise relationships and their corresponding activity category representation vectors are stitched into a relationship matrix $R$ and a category matrix $C$, respectively. Then, the weight matrices $W^Q$, $W^K$ and $W^V$ to be trained are introduced, and the matrix multiplication operations are performed on the weight matrix, the relationship matrix and the category matrix to obtain the matrices *query*, *key* and *value* corresponding to $Q$, $K$ and $V$ and the attention scores as shown in Eqs (2) and (3).

$$Q = R \times W^Q$$
$$K = C \times W^K \quad\quad\quad (2)$$
$$V = C \times W^V$$

$$att(C, R) = Soft \max \left( \frac{QK^T}{\sqrt{d_k}} \right) V \quad\quad\quad (3)$$

where $k$ is the vector dimension of entities and relationships, and $d_k$ is the weight matrix dimension. The higher the attention score obtained by the above attention calculation model, the stronger the relevance of the category $c$ to the relationship $r$, when the category is assigned a higher weight.

## 3.2 Similarity calculation based on contextual information

In traditional blockchain enterprise business activities information, there may be a situation where one entity has similarity with multiple different entities, which is called a "one to many" mapping problem. By introducing contextual information, a contextual information association graph model can be established to more accurately capture the relationships between entities. This model considers the ternary relationship between entities and their related entities, which helps to solve the ambiguity problem in similarity calculation and better reflect the true relationship between entities. In this paper, we introduce enterprise contextual information and build a context structure graph model to calculate the ambiguity of entity name designation. In the inter-block similarity measure, assuming that the total number of blocks in the blockchain is $N$, this paper takes the first block in the blockchain as the block to be compared and the other blocks as the candidate comparison blocks, and uses the average value of the similarity between the block to be compared and the other blocks as the consistency measure of this blockchain, i.e., the average similarity measure.

**3.2.1 Contrasting entity generation.** In this paper, the entity ambiguity calculation is carried out for the example of business entity names and business transaction activities. Based on

Stanford NER for named entity identification, the set of entity designations of the candidate comparison blocks is obtained and denoted as $M = \{m_1, m_2, m_3, \ldots\}$. When performing entity ambiguity calculation, contextual activity information plays an important role as evidence for entity designation. Therefore, the Stanford NER tool is used to extract the activity context information $D$ from the activity information of the removed entity designation set $M$, which constitutes the context set of the activity context information $D$, denoted as $C = \{c_1, c_2, c_3, \ldots\}$.

The named entity designations in the set $M$ are passed through other blocks to obtain the candidate comparison entity set, and the information extraction and formatting of the prepared candidate entity set are sorted out and filtered to generate the final candidate comparison entity set $N_i = \{m_{i1}, m_{i2}, m_{i3}, \ldots\}$.

**3.2.2 Contextual information association graph model construction.** In this paper, the contextual structure in an enterprise entity is represented as an entity-related graph model $G = (V, E)$, where $V$ denotes the vertex set and $E$ denotes the edge set. The construction of the entity-related graph is divided into two steps: vertex set construction and edge set construction.

*(1) Vertex set construction.* Vertex set $V$: the set $D$ of entity contextual activities that appear with a given enterprise activity, where $c_i$ denotes the $i$th contextual information corresponding to this activity, $V = \{c_i \,|\, \forall i, c_i \in C\}$.

Each vertex in the graph is assigned a Confidence Measure (*CM*), which indicates the importance of the node without considering other contextual information, and is calculated as shown in Eq (4).

$$CM(c_i) = \frac{ResultScore(c_i)}{\sum_1^j ResultScore(c_i)} \tag{4}$$

Where *ResultScore*($c_i$) is the matching score obtained based on Google Knowledge Graph, and the higher the matching score, the more accurate the contextual information represents.

*(2) Side set construction.* The edges of the graph model in this paper consist of the path correlations of the contextual information corresponding to this enterprise activity.

In this paper, we propose a new method to determine the path association degree between two messages, i.e., the two-way shortest path determination method, considering the hyperlink structure between messages as a directed graph, remembering that the path from message $A$ link to message $B$ is the forward shortest path, and the path from entity $B$ link to entity $A$ is the backward shortest path, and the shortest path between two nodes is calculated as shown in Eq (5).

$$ShortPath(v_a, v_b) = \frac{FShortPath(v_a, v_b) + BShortPath(v_a, v_b)}{2} \tag{5}$$

Where *FShortPath* denotes the forward shortest path length and *BShortPath* denotes the backward shortest path length. In order to better describe the path correlation between two nodes, the route length is converted into path correlation and calculated as shown in Eq (6).

$$RePath = \frac{1}{1 + ShortPath(v_a, v_b)} \tag{6}$$

From this formula, the smaller the path length between two nodes, the greater the path association between the two nodes. According to the theory of "six degrees of separation", when exploring the shortest path length of two nodes, the upper limit of the shortest path length is set to six, that is, if the path length between two nodes exceeds six, the two nodes are considered to have no path association.

**3.2.3 Inter-block similarity calculation.** In this paper, the semantic similarity is calculated using the name of the business entity and the specific content of the business transaction activities, using *SimText(A,B)* to denote the semantic similarity of the blocks represented by block *A* and block *B*, respectively, and vectorizing the semantic description information represented by block *A* and block *B* as $A = \{m_{11}, m_{12}, m_{13}, \ldots\}$ and $B = \{m_{21}, m_{22}, m_{23}, \ldots\}$, respectively, using cosine similarity calculated as shown in Eq (7):

$$\cos Text(A, B) = \frac{A \cdot B}{\|A\|\|B\|} \tag{7}$$

The normalization process is shown in Eq (8):

$$SimText(A, B) = \frac{\cos Text(A, B) + 1}{2} \tag{8}$$

Finally, the total consistency of the blockchain is calculated using the average measure of the similarity of these blocks, and the average similarity is calculated as shown in Eq (9):

$$CoHerence = \frac{\sum_{i=2}^{N} SimText(A, i)}{N} \tag{9}$$

where *N* is the total number of blocks to be compared, *i* is the blocks other than the blocks to be compared, and *SimText(A,i)* is the similarity measure between the blocks to be compared and the other blocks.

## 3.3 Blockchain quality assessment model construction

In this paper, based on the characteristic attributes of credibility, the credibility of information sources, evaluations and contents of business activities are used to independently characterize the credibility of information, and the results of the characterization are fused.

1. Source-based information trustworthiness characterization. It includes three parts: the trustworthiness of the publishing platform is characterized by the trustworthiness of the information source; the trustworthiness of the page is based on the trustworthiness transfer theory, and the trustworthiness of the current page is characterized by the relevant link evaluation; the trustworthiness of the publisher is taken as the average trustworthiness of the publisher through continuous knowledge accumulation.

2. Comment-based information credibility characterization. Firstly, the comments are classified explicitly and implicitly, then the comment tendency analysis is performed using the lexicon approach, and finally the results of comment-based information credibility characterization are obtained.

3. Content-based credibility characterization. Using the information credibility results derived from using the dictionary of business activity characteristics as the main condition, the information is ranked in terms of credibility and given corresponding weights to smooth out the differences between various types of information, and finally the results of the information credibility characterization based on content characteristics are obtained.

### 3.4 Blockchain content evaluation based on trustworthiness theory

The source or carrier of enterprise activity information directly reflects the degree of information credibility, and this paper characterizes the information credibility by assessing the activity information source.

**3.4.1 Source-based information trustworthiness characterization.** *(1) Credibility of information sources*. In this paper, we assume that the higher the click-through rate of information sources, the higher the credibility. The click through rate is considered an important basis for measuring the credibility of information sources. Firstly, the click through rate reflects the level of interest and recognition of users towards the content of the information source. Users are more likely to click on information they consider useful, authentic, and trustworthy. Therefore, information sources with high click through rates often imply user trust in the source, thus becoming one of the indicators for evaluating credibility. Secondly, click through rate is a direct feedback of user behavior. The behavior of users clicking on a certain information source indicates that they are interested in the content of that source and believe that the content is valuable to them. This positive user behavior feedback is seen as an affirmation of the credibility of the information source. Due to the limitations of certain information sources (e.g., forums, blogs) for access restrictions and their own audience limitations, their click-through rates as a whole will be lower than those of official information sources (e.g., news sites, official websites), so the adverse effects of the above should be smoothed out. The information source credibility model is shown in Eq (10):

$$Site\_R(i) = \frac{hits(i)/\sum_{j=1}^{m} hits(j)}{\sum_{k}^{n} \left( hits(k)/\sum_{j=1}^{m} hits(j) \right)} \tag{10}$$

Where, $Site\_R(i)$ is the credibility of a certain information source, $hits$ denotes the click rate, $i$ denotes the columns related to business activities in the information source, $m$ denotes the number of all columns in the information source, and $n$ is the number of a certain type of information source. The model soothes out the heat by calculating the ratio of column clicks to the whole information source, reducing the rise in clicks on the business activity column due to interest in other information in a source.

*(2) Credibility of information pages*. To a certain extent, the web page where the information is located also characterizes the credibility of the information. In this paper, we propose a method to characterize the trustworthiness of a page based on the information of the relevant links on the page, i.e., the accessibility of the links in the page and the availability of the reached page to judge the trustworthiness of an information source. The specific model is shown in Eq (11):

$$Page\_R(i_{A,B,C}) = \frac{A}{A \cup B} - \frac{C}{A} \tag{11}$$

where $Page\_R(i_{A,B,C})$ denotes the credibility of the information page, $A$ denotes the set of reachable links among the relevant links, $B$ denotes the set of unreachable links among the relevant links, and $C$ denotes the set of unavailable pages among the pages reached by the links. Reachability means that the link can link and open the specified page normally, and availability means that the new page linked to has sufficient information and relevance.

*(3) Credibility of the information publisher*. The credibility of the information publisher is mainly focused on the credibility judgment of the publisher, only for the credibility of the person to judge whether the information is credible, and then gradually converted to credible

information users and untrustworthy information users after a certain knowledge reserve with the deepening of people's understanding of information and the acquisition of knowledge.

In this paper, we consider the network users who can receive information as a whole, and the total number of users is denoted as *N*. Initially, the users who receive data are considered as unknowns *ignorant*; as the information spreads on the network, these users can be further divided into two categories: those who consider the information credible *believed* and those who consider the information untrustworthy *unbelieved*. Based on this, this paper defines a benchmark for measuring the credibility of the information itself, which is used to discern the probability that the publisher of the information releases the information, the probability of determining the truthfulness of the information when it is published.

Therefore, based on the above considerations, the user states in the network can be divided into three categories:

1. *Ignorant*. This state refers to the state in the network when a piece of information has not yet been disseminated; or, when the information is published, users who do not know the truth or falsity of the information and have not yet made a judgment.

2. *Believed*. After having a certain knowledge base, when seeing the information, make a judgment that the information released by the publisher is credible.

3. *Unbelieved*. After having a certain knowledge base, when seeing the information, make a judgment that the information released by the publisher is not credible.

The schematic diagram of state transition between network nodes during information dissemination is shown in Fig 3.

Suppose a publisher publishes a message publicly, then the user who sees the message for the first time is regarded as an unknowing person, and the probability that the message is seen by other users in the unit time starting from moment *t* is α; in contrast, the probability that the user thinks the message is untrustworthy is β.

With the passage of time and the accumulation of their own learned knowledge and awareness, the users' cognitive state of information will change. Therefore, assuming that the probability that a trusted user thinks the information is untrustworthy is γ per unit time

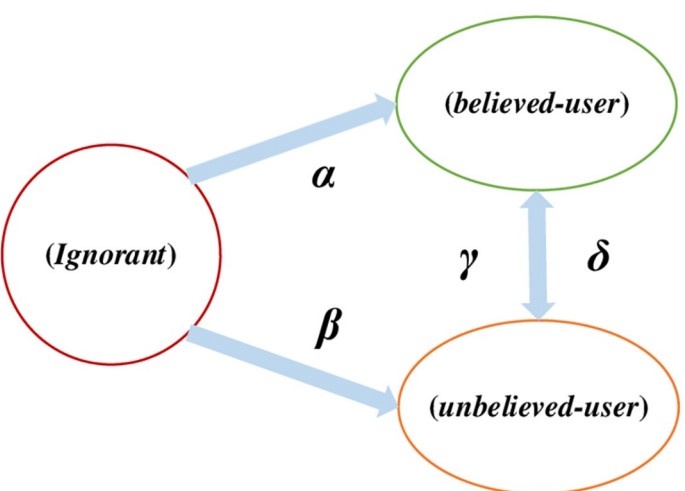

**Fig 3. Inter-node state transition.**

starting from moment *t*, then it is shown that the trusted information user converts to an untrustworthy information user; at the same time, the probability that an untrustworthy information user thinks the information is trustworthy is δ, then the untrustworthy information user converts to a trusted information user. Therefore, based on the above information content-based interaction rules, a system of equations is established as shown in Eq (12).

$$\begin{cases} \dfrac{dI(t)}{dt} = -I(t)[\alpha B(t) + \beta U(t)] \\[2mm] \dfrac{dB(t)}{dt} = \alpha B(t)I(t) - \gamma B(t) + \delta U(t) \\[2mm] \dfrac{dU(t)}{dt} = \beta U(t)I(t) - \delta U(t) + \gamma B(t) \end{cases} \tag{12}$$

Where: $I(t)$, $B(t)$ and $U(t)$ denote the proportion of the total number of users at the time of the unknowns, trusted information users, and untrusted information users, respectively; $I(0)$, $B(0)$ and $U(0)$ denote the value of the number of users in each state in the initial state, and all are non-negative numbers.

In the above interaction rule, assume that the user's trustworthiness of the message publisher is *Publish_R*. When other users see the message, they can directly judge whether the message is trustworthy or not by their intuitive impression of the publisher of the message, then the average trustworthiness probability of the publisher is calculated as shown in Eq (13).

$$Publisher\_R(i) = \frac{1}{N}\sum_{j=1}^{N} Publisher\_R_{j\to i}, i \neq j \tag{13}$$

The probability of implausibility is shown in Eq (14):

$$\bar{T}_i = 1 - Publisher\_R(i) = 1 - \frac{1}{N}\sum_{j=1}^{N} Publisher\_R_{j\to i}, i \neq j \tag{14}$$

*(4) The fusion of credibility characterization results of the source.* The source-based credibility is calculated as shown in Eq (15).

$$R\_source(i) = \sqrt{\alpha Site\_R(i)^2 + \beta Page\_R(i)^2 + \gamma Publisher\_R(i)^2} \tag{15}$$

**3.4.2 Information trustworthiness characterization based on evaluation features.** Information credibility is assessed by calculating the extrinsic representations of the comments, which are first defined as follows.

Invalid comments: Comments that do not contain emotional words or are not related to the credibility of the content of the message.

Explicit comments: Comments that contain words that can clearly characterize the tendency of credibility.

Implicit comments: Comments that contain emotional words but do not have the tendency to explicitly characterize credibility words.

Explicit and implicit comments have different contributions to the evaluation of information credibility based on comments. These two types of comments reveal different attitudes and perspectives of users when facing information. By comprehensively analyzing them, the credibility of information can be more comprehensively evaluated. Explicit comments usually

contain clear attitudes, opinions, or feedback from users. This direct expression helps to understand the user's true view of the information, thereby providing direct clues about the credibility of the information. And Explicit comments often contain more facts, data, or specific information. This clear statement makes it easier to evaluate the accuracy of information and helps confirm whether the information is based on verifiable content. Implicit comments may involve emotional vocabulary, emotional color, or suggestive language, revealing the user's emotional feedback. This is of great significance for evaluating the user experience and impact of information. Implicit comments can provide additional context or information for explicit comments. Sometimes users may not express themselves clearly in their words, but through the tone and emotions in implicit comments, they can gain a deeper understanding of their views. These implicit viewpoints help to gain a more comprehensive understanding of the user's overall perception of information, providing more dimensions for credibility assessment.

1. *Dispositional analysis of explicit commentary representations.* In this paper, the sentiment intensity is adjusted by the modifier window strategy to control the representational value. If the current analyzed word is a sentiment word, the word is placed in the corresponding calculator based on the specific sentiment tendency, and the degree to which the current sentiment word is modified by the modifier words in the sliding window is taken out to obtain the value of the word's contribution to the sentiment of the sentence. If the sliding window contains a negative word when the current sentiment word is analyzed, the sentiment tendency is reversed and the sliding window range is used to control the range of action of the negative word. Finally, the sentiment tendency is calculated for the whole comment tendency, which is calculated as shown in Eq (16).

$$R\_review\_dominant = \sum_{i=1}^{N} \sum \frac{R - D}{L} \tag{16}$$

where *R_review_dominant* denotes the value of a user's comment on the credibility of the message, *N* denotes the number of sentences in the total comment, *R* denotes the score of credible representations in a sentence, *D* denotes the score of suspicious representations in a sentence, and *L* denotes the length of a sentence. When the single-sentence sentiment score lies in the (- 0.1, + 0.1) interval, the comment has a weak tendency to be characterized and is not considered.

2. *Implicit commentary representation propensity analysis.* The analysis of representational tendency for implicit comments is performed in two steps: first, the relevance of the comments to the message is calculated; second, the representational tendency value is calculated for the relevant implicit comments.

   1. *Relevance of comments to information.* In this paper, a probabilistic retrieval model is used to calculate the relevance of information and comments, i.e., comments and information are considered as retrieval problems. The set of information constitutes the set of documents when the comment *C* is assumed to be a query string and the information *I* is treated as a document. On the basis of the classical probabilistic retrieval model, it is smoothed as shown in Eq (17).

$$P(CI) = \lambda \prod_{w \in C} p(wI) + (1 - \lambda) \prod_{w \in C} \sum_{t \in t_1} \prod_{w \in t} p(w|t) * p(t|I) \tag{17}$$

where *I* is the message, *C* is an entry in the set of comments of *I*, *w* is a word in *C*, $p(C|I)$ is the probability of *I* generating *C*, and $p(w|I)$ is the probability of *w* appearing in *I*. $t_I$ is the set of topics of message *I*, *t* is a topic in $t_I$, λ is a parameter, $p(w|t)$ is the probability of

word $w$ appearing in topic $t$, and $p(t|I)$ is the probability of topic $t$ appearing in message $I$. $p(C|I)$ is calculated for all comments in message $I$. If the probability value is greater than a certain threshold, the relevant comment is determined to be relevant to the message.

2. *Related implicit commentary on representational tendencies.* In addition to explicit emotion words, some implicit words also have attitudinal characteristics, and for implicit evaluation representation tendency values are calculated as shown in Eq (18).

$$R\_review\_do \min ant(R) = \sum_i (1 + p(a_i)/count(a|R)) \times w(a_i) \qquad (18)$$

where $a$ denotes a paragraph, sentence or word, and $R$ corresponds to a whole comment, paragraph or sentence, the sentiment score of $R$ is denoted as $w(R)$, $p(a_i)$ denotes the position of the sentiment unit $a_i$ in $R$, and $count(a|R)$ denotes the total number of $R$ containing $a$.

3. *Comment-based fusion of credibility characterization results.* The process of fusion is divided into two steps: first, the objects of the comments are obtained and classified; second, the results of the classified comments are fused in the way shown in Eq (19).

$$R\_review = \sum_{(i,j)} Tar(R_i) * R_j + \sum_k R_k \qquad (19)$$

where $R\_review$ is the trustworthiness of the message based on the comment, $Tar(R_i)$ is the characterization of the trustworthiness of the message by the object $i$ being commented on, $R_j$ is the characterization of the trustworthiness of a comment $j$ on $i$, and $R_k$ is the characterization of the trustworthiness of a direct comment on the message.

**3.4.3 Information trustworthiness characterization based on content features.** Existing methods of content credibility calculation assess the content of a certain type of activity information with the help of external information and lack a direct and reliable evaluation algorithm for information from multiple sources. This paper proposes a method for characterizing information trustworthiness using data content features based on a study of enterprise activity information characteristics, which is based on two assumptions:

1. Feature words determine the professionalism of information, i.e. the more words characterizing enterprise activities in the information content, the more professional the information is.

2. The professionalism of the information determines the credibility of the information, i.e. the more professional the information is, the more credible it is.

Considering the uneven distribution of corporate activity information features in information from multiple sources, this paper uses a corporate activity feature dictionary to count the frequency of corporate activity feature words in various types of information. The credibility of the content is calculated as shown in Eq (20).

$$R\_content(i) = \left( \frac{vword\_all}{\sum web\_word} \right) * \frac{\alpha \sum word\_f + \beta \sum word\_s + \gamma \sum word\_t}{\mu word\_all} \qquad (20)$$

where *word_f*, *word_s* and *word_t* denote the number of common, characteristic and specialized words occurring in the message, respectively, α, β and λ denote the associated weights, and μ is the information type characteristic coefficient.

Finally, information trustworthiness in the blockchain is a combined measure of source-based trustworthiness, comment-based trustworthiness and content-based trustworthiness, as shown in Eq (21).

$$Accu = \alpha R\_source(i) + \beta R\_review + \gamma R\_content(i) \tag{21}$$

Among them $\alpha < \gamma < \beta$, since the information source is the key factor to measure the credibility of information, such as official information is 100% credible, the credibility of the information source has the largest weight, the content credibility is the second, and the comment-based credibility has the smallest weight since the comment depends on the subjectivity of the commenter. Some situation may affect the method like different fields may have different professional standards and vocabulary usage habits. Some fields may place more emphasis on the use of professional terminology, while others may place more emphasis on common language. Therefore, the relationship between the number of feature words and professionalism may vary between different fields. Sometimes, the number of feature words does not always reflect the depth and detail level of information. Some content may require more professional terminology and detailed explanations, which may not necessarily lead to an increase in the number of feature words. Therefore, quantity cannot fully represent the professionalism and credibility of the text. But in most cases, these issues will not have a significant impact on the conclusions of the model output, so no additional discussion is needed.

## 3.5 Blockchain value assessment based on the amount of information

To measure the validity of information on business activities in the blockchain, this paper adopts the method of calculating the amount of information, which is measured by the amount of information on business activities in the block. The method realizes the value assessment of the blockchain through the amount of information contained in the blockchain, determines the indeterminacy of the information representation source based on the determination of the indeterminacy of the information selection of the source, and transforms the study of the indeterminacy of a random event into the study of the totality of the probability corresponding to that random event. Its relevant definitions and properties are as follows:

**Definition** Self-informative amount. The self-information of an arbitrary random event is defined as the negative of the logarithm of the probability of occurrence of the event. Let the probability of the event $x_i$ be $p(x_i)$. Then its self-information is defined as shown in Eq (22).

$$I(x_i) = -\log p(x_i) \tag{22}$$

**Property 1** There is no negative value for the amount of information, and the smaller the value means the smaller the amount of information carried, and the larger the value, the greater the amount of information.

**Property 2** The amount of information is additive in nature. According to this property, the amount of information in a blockchain node can be considered as the sum of the amount of information of each active message in the block.

Information amount can quantitatively describe exactly how much information about business activities is contained in each block in the blockchain. In the process of blockchain information amount calculation, information amount represents the amount of information contained in a block, and is a measure of the validity of activity information. Each block can be regarded as a discrete source, and if it is a discrete random variable (a certain chain), the set of values of the random variable and its probability measure are as shown in Eq (23).

$$X = \{x_1, x_2, \ldots, x_n\}, \; p_i = P[X = x_i] \tag{23}$$

The probability space of the discrete random variable $X$ is shown in Eq (24).

$$\begin{bmatrix} X \\ p(x) \end{bmatrix} = \begin{bmatrix} x_1, x_2, \cdots, x_i, \cdots, x_n \\ p_1, p_2, \cdots, p_i, \cdots, p_n \end{bmatrix}, \ \sum_{i=1}^{n} p_i = 1 \tag{24}$$

where $p_i$ is the probability of occurrence of active information in block $x_i$. The amount of information is calculated as shown in Eq (25).

$$Validity = -\sum_{i=1}^{n} p(x)\log_2 p(x), \ p(x) = P(X = x) \tag{25}$$

Here, $X$ is a random variable, *Validity* is the amount of information contained in the block, and $p(x)$ is the probability distribution function of the variable $X$.

In practical applications, the use of this method is more practical for unknown unordered structures. The degree of information uncertainty is expressed by a non-negative value, and the magnitude of the information quantity value directly determines the amount of information. For a certain blockchain, the amount of information can be measured by this value, and the larger the value, the more information the blockchain contains, and thus the greater the value of the blockchain.

## 3.6 Quality assessment model

Multi-source heterogeneous blockchain quality assessment is used to assess the consistency, trustworthiness, and value of the information on business activities stored in the blockchain. In this paper, we propose a weighted approach to construct a blockchain quality assessment model, which uses the similarity of block representations, source-based block trustworthiness representations, and the weighted average of the values contained in the blocks to measure the comprehensive quality of the blockchain. The final evaluation model is shown in Eq (26).

$$Q = \alpha CoHerence + \beta Accu + \gamma Validity \tag{26}$$

where $\alpha$, $\beta$ and $\gamma$ represent the weights of three assessment metrics, respectively. Since blockchain value is an important indicator of blockchain quality, the greater the value contained in the blockchain, the greater the application value of that blockchain, followed by information trustworthiness, hence the weight $\gamma > \beta > \alpha$.

## 3.7 Case analysis of enterprise alliance chain evaluation

This section mainly evaluates the quality of our existing enterprise blockchain business activities, and obtains the final blockchain data quality evaluation result through comprehensive consistency, accuracy, and value calculation results. We will use effective data obtained from local enterprises to apply in this section.

Firstly, based on the credibility of the information source, it is mainly measured comprehensively by the credibility of the information source, information page, and publisher. For the credibility of the information source, its click through rate is calculated as click through rate = click through rate/exposure rate (as used in the example in this article); For the credibility of information pages, it is mainly measured by the accessibility of the page where the information is located and whether it is available after accessibility, while the credibility of publishers is judged by users through certain knowledge accumulation. Based on the calculation results of the above three factors, the credibility of the information source can be

calculated as follows:

$$R\_source = \sqrt{0.6*\left(\frac{1}{20\%}\right)^2 + 0.2*0.75^2 + 0.2*\left(\frac{11}{40}\right)^2} = 3.9 \tag{27}$$

Secondly, calculate the credibility of the information evaluation. When the evaluation of the activity event is none, the evaluation characterizes the credibility of the information as 0; When the evaluation of an activity event is an explicit evaluation, its credibility representation value is directly calculated; When the evaluation of an activity event is calculated as an implicit evaluation, the first step is to determine whether the evaluation is related to the information, and then calculate the tendency value of the relevant representation. The above blockchain calculates the credibility representation values based on evaluation information as follows:

$$R\_review = \sum_{i=1}^{2} \sum \frac{0-0}{2} + \frac{2-1.5}{2} = 0.25 \tag{28}$$

The representation of information credibility based on content features is mainly measured by the number of common words, feature words, and professional words in the information content. By conducting relevant statistics on the frequency of various features in enterprise business activity information and calculating them:

$$R_{\text{content}} = \frac{0.2*38 + 0.3*12 + 0.5*14}{0.3*64} * \left(\frac{0.2*64}{500}\right) = 0.01598 \tag{29}$$

The credibility of information sources is determined by a combination of source based, evaluation based, and content-based credibility. The calculated credibility of information sources is as follows:

$$\text{Accu}(A) = 0.5*3.9 + 0.2*0.25 + 0.3*0.01598 = 2.005 \tag{30}$$

Finally, calculate the value information of blockchain:

$$\text{Validity} = -\left(\frac{3}{8}\log_2\frac{3}{8} + \frac{1}{8}\log_2\frac{1}{8} + \frac{1}{2}\log_2\frac{1}{2} + \frac{1}{2}\log_2\frac{1}{2} + \frac{1}{4}\log_2\frac{1}{4} + \frac{1}{8}\log_2\frac{1}{8}\right) = 2.782 \tag{31}$$

The final quality evaluation result of blockchain is weighted by three factors:

$$Q = 0.1*0.72 + 0.2*2.005 + 0.7*2.782 = 2.4204 \tag{32}$$

The above process can prove that the final quality evaluation model of blockchain in this article has good operability and feasibility.

## 4 Experiments and results

### 4.1 Experiments setting

The experimental hardware and software environment is shown in Table 1.

The experimental dataset consists of two parts, which are official data searched and crawled on the search engine with the subject term of business activities and real data of enterprises, totaling about 210,000 items, and then simple data cleaning is performed by filling in missing values, smoothing or removing outlier points and other operations to get about 200,000 items of data, and the details of the dataset are listed in Table 2. The datasets are available from https://doi.org/10.6084/m9.figshare.25143503.v1. In this paper, simulation experiments are conducted in terms of model parameter taking and model evaluation efficiency respectively,

**Table 1. Experimental platform sets.**

| Hardware | Bytes |
|---|---|
| CPU | Intel Core i5-7400 |
| CPU running | 3GHz |
| Memory | 8GB |
| Operation system | Windows11 |
| Programming environment | PyCharm 2020.3.3 |
| Programming language | Python |
| Programming language version | 3.7.3 |

and the model built in this paper (DQAM) is used to compare with other models, and the comparison models are AHP, DSMM, etc.

## 4.2 Model parameter setting

The experiments in this section focus on verifying the parameter settings of the proposed evaluation model, through which it can be seen that the blockchain value takes the largest weight, followed by the information credibility, i.e., weight $\gamma > \beta > \alpha$. The values of the three can affect the model evaluation effect. Therefore, the experiments in this section verify three different parameter settings and then select the optimal weight setting. As shown in Fig 4, the better relative running efficiency of the vertical axis indicates the better setting of this parameter, and the horizontal axis indicates the number of data sets run, in units of ten thousand, and the effect of different $\gamma$: $\beta$: $\alpha$ ratio runs is as follows.

As can be seen from Fig 4, when the model parameters are set to $\gamma$: $\beta$: $\alpha$ = 7: 2: 1, the model runs more efficiently, optimizing 5% to 10% over the other ratios.

## 4.3 Consistency assessment efficiency comparison

In this section, the inconsistency detection efficiency of entity information is evaluated on the dataset in Table 2, where the horizontal axis represents the number of data entries, in units of ten thousand, and the vertical axis represents the consistency assessment efficiency, and the efficiency is compared by the structured gradient tree boosting (SGTB) algorithm, disambiguation of inconsistent record pairs based on factor graphs (DIBFM), neural network disambiguation with multi-view concerns (NDMP) method and the semantic similarity calculation method between blocks based on contextual information (SCBCI) proposed in this paper, and the experimental results are shown in Fig 5.

**Table 2. Experimental data sets.**

| Entity Name | Bytes |
|---|---|
| Name | 16bytes |
| Type | 8 bytes |
| activityOntology | Variable length bytes |
| comment | Variable length bytes |
| informationSource | 16 bytes |
| Code | 4 bytes |
| registrationDay | 4 bytes |
| capital | 16 bytes |
| address | 32 bytes |

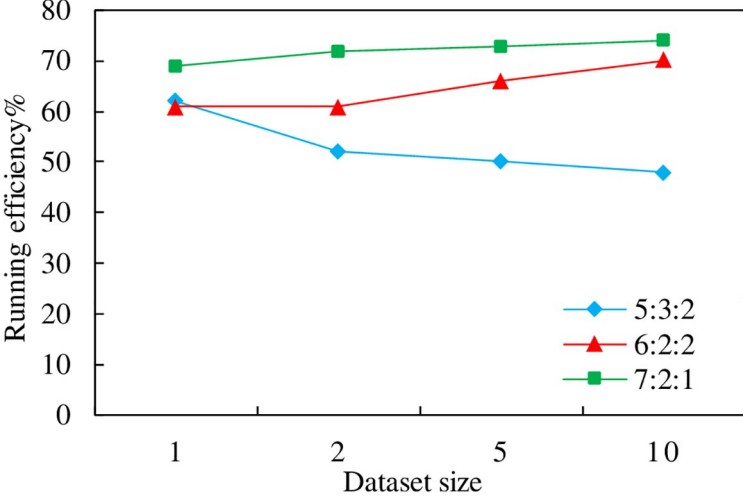

**Fig 4. Model parameter settings.**

As can be seen from Fig 5, with the increase of data amount, the evaluation efficiency of the consistency assessment method proposed in this paper is gradually improved compared with the existing methods, and the efficiency is not reduced due to the large amount of data.

## 4.4 Accuracy assessment efficiency comparison

This section validates the efficiency of the trustworthiness theory-based blockchain content evaluation (CEBT) method. The vertical axis is the operational efficiency, and the horizontal axis represents the number of data entries, in units of ten thousand, and the specific operational efficiency comparison of each method on different data sets is shown in Fig 6. As can be seen from the figure, the average evaluation efficiency of the CEBT method is higher than that of the other methods. The comparison methods are the sensor data trustworthiness assessment

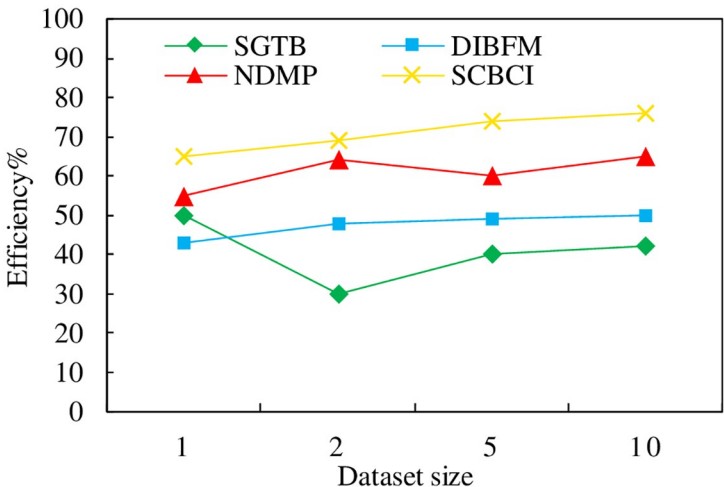

**Fig 5. Consistency assessment efficiency comparison.**

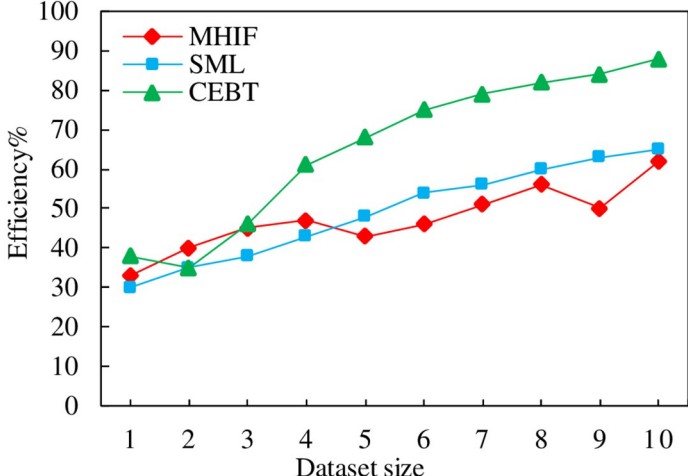

**Fig 6. Accuracy assessment efficiency comparison.**

(MHIF) method based on multi-source heterogeneous information fusion and the supervised machine learning (SML) method for user-generated content trustworthiness assessment.

## 4.5 Comparison of value assessment efficiency

The experiments were conducted by simulating the comparison of the value assessment efficiency using the amount of information approach (VIS) with the value assessment approach based on data distribution (VADD) and the VW&ICM calculation model, with the horizontal coordinate indicating the amount of data, in units of ten thousand, and the vertical coordinate indicating the assessment efficiency, and the experimental results are shown in Fig 7.

From the experimental results, it can be found that the information amount-based assessment method proposed in this paper is more efficient. The main reason for this is that the information-amount-based data value assessment focuses on the data value itself rather than

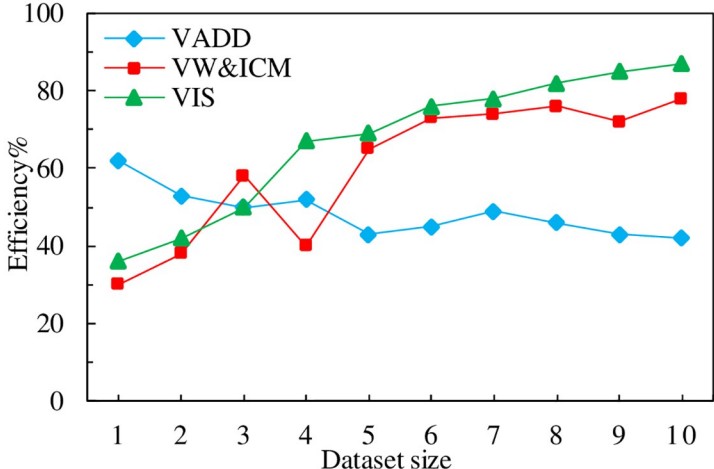

**Fig 7. Valuation efficiency comparison.**

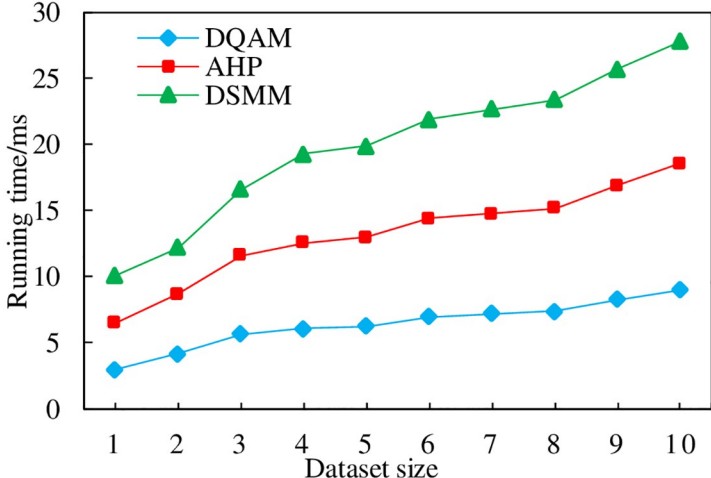

**Fig 8. Model efficiency comparison.**

its secondary factors such as distribution and regularity, so the method is intuitive, clear and more efficient.

## 4.6 Total evaluation efficiency comparison

This section compares the running efficiency of each model, the horizontal coordinate represents the entity data set in the blockchain, in units of ten thousand, the vertical coordinate indicates the time required to run each model, and the experimental results are shown in Fig 8.

From the results, it can be seen that the average running time of AHP and DSMM methods is around 10~15ms, and the evaluation time of DQAM model proposed in this paper is around 5ms, and the evaluation time of DQAM model basically fluctuates little with the increase of data amount, and the average time is the shortest compared with other methods, which is more suitable for users to conduct blockchain quality evaluation.

## 5 Conclusion

This paper proposes a multi-source heterogeneous blockchain data quality assessment model for enterprise business activities. Initially, leveraging pertinent enterprise entity data within the blockchain, we introduce a triadic representation of enterprise entities utilizing the CEKGRL model, intricately linked with relevant activity categories. This integration enables the construction of an association graph model for entity context information, thus enhancing the efficiency of similarity computations between blocks. Subsequently, through an examination of blockchain content, we ascertain the credibility of blockchain information via a credibility characterization method. This method is grounded in information source analysis, information commentary evaluation, and the content of business activity information, amalgamating all characterization outcomes to provide a comprehensive credibility assessment. Finally, the value assessment method is used to evaluate the value amount contained in the content of the blockchain, and the above information is combined to obtain the blockchain quality assessment model.

This paper has conducted in-depth research on the blockchain data quality evaluation model, proposed corresponding solutions for the shortcomings of current evaluation methods, and achieved some meaningful research results. However, the application of blockchain

technology involves other fields besides the financial field. For its different application scenarios, the following issues related to this study need further exploration and research:

1. The blockchain data quality evaluation model in this paper is aimed at the business activities data information of enterprises in the financial field, while other fields are also producing a lot of data information at all times. Therefore, applying the evaluation model proposed in this paper to data in other fields is the next research direction.

2. The model proposed in this paper is based on the established blockchain, but in reality, the information in the blockchain is sometimes stored dynamically, and may be linked at any time. Therefore, when new block information is generated in the blockchain, how to evaluate its quality in real time is worth studying.

## Acknowledgments

The authors thank the reviewers for their constructive comments in improving the quality of this paper.

## Preprint version statement

A preprint version of the paper has been published in Research Square, Zhang Haolin, Su Li, and Likuan Du making revisions to the second edition, so the order of authors has been adjusted and two new authors have been added.

## Author Contributions

**Conceptualization:** Su Li, Junlu Wang.

**Data curation:** Su Li, Junlu Wang.

**Formal analysis:** Su Li, Baoyan Song.

**Funding acquisition:** Baoyan Song.

**Investigation:** Ran Zhang.

**Methodology:** Ran Zhang.

**Project administration:** Ran Zhang.

**Resources:** Likuan Du.

**Software:** Likuan Du.

**Supervision:** Likuan Du.

**Visualization:** Haolin Zhang, Wanting Ji.

**Writing – original draft:** Haolin Zhang, Wanting Ji, Junlu Wang.

**Writing – review & editing:** Haolin Zhang, Wanting Ji, Junlu Wang.

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
