## [Decision Letter · Decision Letter 0]

2 Jan 2024

PONE-D-23-32614Multi-source heterogeneous blockchain data quality assessment model for enterprise business activities

PLOS ONE

Dear Dr. Wang,

Thank you for submitting your manuscript to PLOS ONE. After careful consideration, we feel that it has merit but does not fully meet PLOS ONE’s publication criteria as it currently stands. Therefore, we invite you to submit a revised version of the manuscript that addresses the points raised during the review process.

We look forward to receiving your revised manuscript.

Kind regards,

Asadullah Shaikh, Ph.D.

Academic Editor

PLOS ONE

2. In your Methods section, please include additional information about your dataset and ensure that you have included a statement specifying whether the collection and analysis method complied with the terms and conditions for the source of the data.

“This study was supported by the Applied Basic Research Program of Liaoning Province(No.2022JH2/101300250); the Digital Liao-ning Smart Building Strong Province (Direction of Digital Economy)(No.13031307053000568); the National Key R&D Program of China(No.2021YFF0901004); the Central Government Guides Local Science and Technology Development Foundation Project of Liaoning Province (No.2022JH6/100100032); the Natural Science Foundation of Liaoning Province (No.2022-KF-13-06).”

5. In this instance it seems there may be acceptable restrictions in place that prevent the public sharing of your minimal data. However, in line with our goal of ensuring long-term data availability to all interested researchers, PLOS’ Data Policy states that authors cannot be the sole named individuals responsible for ensuring data access (http://journals.plos.org/plosone/s/data-availability#loc-acceptable-data-sharing-methods).

6. PLOS requires an ORCID iD for the corresponding author in Editorial Manager on papers submitted after December 6th, 2016. Please ensure that you have an ORCID iD and that it is validated in Editorial Manager. To do this, go to ‘Update my Information’ (in the upper left-hand corner of the main menu), and click on the Fetch/Validate link next to the ORCID field. This will take you to the ORCID site and allow you to create a new iD or authenticate a pre-existing iD in Editorial Manager. Please see the following video for instructions on linking an ORCID iD to your Editorial Manager account: https://www.youtube.com/watch?v=_xcclfuvtxQ.

Reviewers' comments:

Reviewer's Responses to Questions

**Comments to the Author**

1. Is the manuscript technically sound, and do the data support the conclusions?

Reviewer #1: Yes

Reviewer #2: Yes

2. Has the statistical analysis been performed appropriately and rigorously? 

Reviewer #1: Yes

Reviewer #2: Yes

3. Have the authors made all data underlying the findings in their manuscript fully available?

Reviewer #1: Yes

Reviewer #2: Yes

4. Is the manuscript presented in an intelligible fashion and written in standard English?

Reviewer #1: Yes

Reviewer #2: Yes

5. Review Comments to the Author

Reviewer #1: Dear Author,

I have carefully reviewed the manuscript titled "Multi-Source Heterogeneous Blockchain Data Quality Assessment for Enterprise Business Activities." While the paper proposes an intriguing approach to address the challenges in evaluating the consistency, credibility, and value of information in enterprise business activities stored in blockchain, there are several aspects that need clarification and improvement:

Experimental Details and Data:

The manuscript lacks specific details on experimental setups, controls, replications, and sample sizes. To ensure the technical soundness of your research, it is essential to provide comprehensive information about the experiments conducted, including the datasets used, methodology for control, and the size of the samples involved.

Clarity in Methodology:

The methodology section, especially in Section 3.1 regarding the CEKGRL model, is intricate and could benefit from additional clarity. A more detailed explanation of the entity triad representation, attention mechanism, and how the model addresses inconsistencies in entity information would enhance the reader's understanding.

Contextual Information and Ambiguity:

While the introduction of contextual information for similarity calculation is interesting, further details are needed to understand how this addresses the "one-to-many" mapping problem and ambiguity in entity name designation. A clearer explanation of how the contextual information association graph model contributes to resolving these issues would be beneficial.

Source Credibility Model:

The credibility characterization of information sources in Section 3.4 requires more explanation, particularly regarding the calculation of the click-through rate and how it directly correlates with credibility. Please provide more insights into how this model aligns with established credibility standards.

Comment-Based Information Credibility:

The methodology for comment-based information credibility characterization (Section 3.4) could be further detailed. Consider providing examples or case studies to illustrate how explicit and implicit comments are classified and contribute to the overall credibility assessment.

Content-Based Credibility Characterization:

The content-based credibility characterization assumes a direct relationship between the number of feature words and professionalism/credibility. It is important to discuss potential limitations of this assumption and consider other factors that may influence credibility.

Conclusion and Implications:

The conclusion should summarize the key findings and their implications. Discuss the potential limitations of the proposed method and suggest avenues for future research to strengthen the manuscript's overall impact.

I appreciate the effort invested in this research and believe that addressing these concerns will significantly enhance the manuscript's quality and contribution to the field.

Best regards,

Reviewer #2: Start by summarizing the paper's objectives, which focus on assessing the quality of data in multi-source heterogeneous blockchain environments, particularly in the context of enterprise business activities. Highlight the importance of this topic in the rapidly evolving blockchain landscape and its relevance to businesses relying on blockchain technology.

1. The abstract needs refinement. Clearly articulate the role of the proposed study and highlight the actual contributions. This will provide readers with a better understanding of the paper's significance.

2. Expand the introduction to provide more context and background information. A more detailed introduction will better set the stage for the problem you are addressing, helping readers grasp the significance of your research.

3. Include a section that compares your work with previous studies and techniques. This comparison can provide additional support for the validity and novelty of your research. Highlight similarities, differences, and advancements over prior work

4. Suggest a more in-depth analysis of security and privacy concerns associated with multi-source blockchain data.

Encourage the inclusion of case studies or practical examples to demonstrate the model's application in real business scenarios and also Propose collaborations with blockchain enterprises for real-world testing and validation of the model.

6. PLOS authors have the option to publish the peer review history of their article (what does this mean?). If published, this will include your full peer review and any attached files.

Reviewer #1: No

Reviewer #2: **Yes: **Tahir Alyas

---

## [Author Response · Author response to Decision Letter 0]

15 Feb 2024

Dear Editor and Reviewers,

Thanks for your comments concerning our manuscript entitled “multi-source heterogeneous blockchain data quality assessment model for enterprise business activities". The comments are very valuable and helpful for revising and improving our manuscript. We have carefully revised the manuscript and responded all comments below. The changes will not affect the content and framework of the paper. The revised sentences are highlighted in yellow in the manuscript.

Best regards,

Baoyan Song

Reviewer 1:

 Experimental Details and Data. The manuscript lacks specific details on experimental setups, controls, replications, and sample sizes. To ensure the technical soundness of your research, it is essential to provide comprehensive information about the experiments conducted, including the datasets used, methodology for control, and the size of the samples involved.

Thanks for the reviewer 's advice. We have added a separate subsection to the experimental section that describes the experimental setting and experimental dataset of this paper, as well as other models that were tested in comparison with the model proposed in this paper. The details are as follows.

Experiments setting

The experimental hardware and software environment is shown in Table 1.

Table 1 Experimental platform sets

Hardware Bytes

CPU Intel Core i5-7400

CPU running 3GHz

Memory 8GB

Operation system Windows11

Programming environment PyCharm 2020.3.3

Programming language Python

Programming language version 3.7.3

The experimental dataset is searched on search engines with the subject term of business activities and crawled official data, about 110,000 items, and then simple data cleaning is performed by filling in missing values, smoothing or removing outlier points and other operations to get about 100,000 items of data, and the details of the dataset are listed in Table 2. In this paper, simulation experiments are conducted in terms of model parameter taking and model evaluation efficiency respectively, and the model built in this paper (DQAM) is used to compare with other models, and the comparison models are AHP, DSMM, etc.

Table 2 Experimental data sets

Entity Name Bytes

name 16bytes

type 8 bytes

activityOntology Variable length bytes

comment Variable length bytes

informationSource 16 bytes

code 4 bytes

registrationDay 4 bytes

capital 16 bytes

address 32 bytes

 Clarity in Methodology. The methodology section, especially in Section 3.1 regarding the CEKGRL model, is intricate and could benefit from additional clarity. A more detailed explanation of the entity triad representation, attention mechanism, and how the model addresses inconsistencies in entity information would enhance the reader's understanding.

Thanks for the reviewer 's advice. We have reorganized the logic of Section 3.1 and added additional explanations related to the entity triad representation, the attention mechanism, and how the model resolves inconsistencies in entity information. The details are as follows.

The triple structure is a data model widely used for representing information, consisting of three components: subject, predicate, and object. It is extensively applied in fields such as knowledge graphs and the semantic web. The triple structure is highly flexible and can be employed to represent various types of information. Different combinations of subjects, predicates, and objects can capture rich semantic relationships, making it suitable for diverse knowledge representation. In summary, the advantages of the triple structure lie in its flexibility, clear semantic representation, scalability, and effective expression of relationships, making it a key technology in the fields of knowledge representation and the semantic web.

 Contextual Information and Ambiguity. While the introduction of contextual information for similarity calculation is interesting, further details are needed to understand how this addresses the "one-to-many" mapping problem and ambiguity in entity name designation. A clearer explanation of how the contextual information association graph model contributes to resolving these issues would be beneficial.

Thanks for the reviewer 's advice. We have added a description of contextual information and ambiguity in Section 3.2. We explain specifically how contextual information solves the "one-to-many" mapping problem and the ambiguity of entity name designation.

In traditional blockchain enterprise business activities information, there may be a situation where one entity has similarity with multiple different entities, which is called a "one to many" mapping problem. By introducing contextual information, a contextual information association graph model can be established to more accurately capture the relationships between entities.

 Source Credibility Model. The credibility characterization of information sources in Section 3.4 requires more explanation, particularly regarding the calculation of the click-through rate and how it directly correlates with credibility. Please provide more insights into how this model aligns with established credibility standards.

Thanks for the reviewer 's advice. We have added a description related to the information source credibility model in section 3.4, explaining the process of calculating the click-through rate and how it relates to credibility. We also explain the way in which the model is aligned with existing credibility standards. The details are presented below.

The click through rate is considered an important basis for measuring the credibility of information sources. Firstly, the click through rate reflects the level of interest and recognition of users towards the content of the information source. Users are more likely to click on information they consider useful, authentic, and trustworthy. Therefore, information sources with high click through rates often imply user trust in the source, thus becoming one of the indicators for evaluating credibility. Secondly, click through rate is a direct feedback of user behavior. The behavior of users clicking on a certain information source indicates that they are interested in the content of that source and believe that the content is valuable to them. This positive user behavior feedback is seen as an affirmation of the credibility of the information source.

 Comment-Based Information Credibility. The methodology for comment-based information credibility characterization (Section 3.4) could be further detailed. Consider providing examples or case studies to illustrate how explicit and implicit comments are classified and contribute to the overall credibility assessment.

Thanks for the reviewer 's advice. We have added a related description of comment-based information credibility in Section 3.4, with example explanations that illustrate how explicit and implicit comments can be categorized and how they can contribute to the overall credibility assessment. The details are as follows.

Explicit and implicit comments have different contributions to the evaluation of information credibility based on comments. These two types of comments reveal different attitudes and perspectives of users when facing information. By comprehensively analyzing them, the credibility of information can be more comprehensively evaluated. Explicit comments usually contain clear attitudes, opinions, or feedback from users. This direct expression helps to understand the user's true view of the information, thereby providing direct clues about the credibility of the information. And Explicit comments often contain more facts, data, or specific in-formation. This clear statement makes it easier to evaluate the accuracy of information and helps confirm whether the information is based on verifiable content. Implicit comments may involve emotional vocabulary, emotional color, or suggestive language, revealing the user's emotional feedback. This is of great significance for evaluating the user experience and impact of information. Implicit comments can provide additional con-text or information for explicit comments. Sometimes users may not express themselves clearly in their words, but through the tone and emotions in implicit comments, they can gain a deeper understanding of their views. These implicit viewpoints help to gain a more comprehensive understanding of the user's overall perception of information, providing more dimensions for credibility assessment.

 Content-Based Credibility Characterization. The content-based credibility characterization assumes a direct relationship between the number of feature words and professionalism/credibility. It is important to discuss potential limitations of this assumption and consider other factors that may influence credibility.

Thanks for the reviewer 's advice. We have added a relevant description of the content credibility characteristics in the last paragraph of Section 3.4. The details are as follows.

Some situation may affect the method like different fields may have different professional standards and vocabulary usage habits. Some fields may place more emphasis on the use of professional terminology, while others may place more emphasis on common language. Therefore, the relationship between the number of feature words and professionalism may vary between different fields. Sometimes, the number of feature words does not always reflect the depth and detail level of information. Some content may require more professional terminology and detailed explanations, which may not necessarily lead to an increase in the number of feature words. Therefore, quantity cannot fully represent the professionalism and credibility of the text. But in most cases, these issues will not have a significant impact on the conclusions of the model output, so no additional discussion is needed.

 Conclusion and Implications. The conclusion should summarize the key findings and their implications. Discuss the potential limitations of the proposed method and suggest avenues for future research to strengthen the manuscript's overall impact.

Thanks for the reviewer 's advice. We have reorganized the conclusion section and added a description related to the limitations of the paper's methodology and future perspectives as follows.

This paper has conducted in-depth research on the blockchain data quality evaluation model, proposed corresponding solutions for the shortcomings of current evaluation methods, and achieved some meaningful research results. However, the application of blockchain technology involves other fields besides the financial field. For its different application scenarios, the following issues related to this study need further exploration and research:

(1) The blockchain data quality evaluation model in this paper is aimed at the business activities data in-formation of enterprises in the financial field, while other fields are also producing a lot of data information at all times. Therefore, applying the evaluation model proposed in this paper to data in other fields is the next research direction.

(2) The model proposed in this paper is based on the established blockchain, but in reality, the information in the blockchain is sometimes stored dynamically, and may be linked at any time. Therefore, when new block information is generated in the blockchain, how to evaluate its quality in real time is worth studying.

Reviewer 2: 

 The abstract needs refinement. Clearly articulate the role of the proposed study and highlight the actual contributions. This will provide readers with a better understanding of the paper's significance.

Thanks for the reviewer 's advice. We have refined the abstract by adding a relevant description of the problem and significance of the paper to be studied as follows.

Blockchain-based applications are becoming more and more widespread in business operations. In view of the shortcomings of existing enterprise blockchain evaluation methods, this paper proposes a multi-source heterogeneous blockchain data quality evaluation model for enterprise business activities, so as to achieve efficient evaluation of business activity information consistency, credibility and value. This paper proposes a multi-source heterogeneous blockchain data quality assessment method for enterprise business activities, aiming at the problems that most of the data in enterprise business activities come from different data sources, information representation is inconsistent, information ambiguity between the same block chain is serious, and it is difficult to evaluate the consistency, credibility and value of information. The method firstly proposes an entity information representation method based on the Representation learning for fusing entity category information (CEKGRL) model, which introduces the triad structure of related entities in blockchain, then associates them with enterprise business activity categories, and carries out similarity calculation through contextual information to achieve blockchain information consistency assessment. After that, a trustworthiness characterization method is proposed based on information sources, information comments, and information contents, to obtain the trustworthiness assessment of the business. Finally, based on the information trustworthiness characterization, a value assessment method is introduced to assess the total value of business activity information in the blockchain, and a blockchain quality assessment model is constructed. The experimental results show that the proposed model has great advantages over existing methods in assessing inter-block consistency, intra-block activity information trustworthiness and the value of blockchain.

 Expand the introduction to provide more context and background information. A more detailed introduction will better set the stage for the problem you are addressing, helping readers grasp the significance of your research.

Thanks for the reviewer 's advice. We have added a description in the third paragraph of the introduction of the Section 1 to illustrate some of the problems of traditional enterprise blockchain, to better introduce the background information of the problem to be researched in this paper, and to highlight the significance of the research in this paper. The details are as follows.

In the enterprise blockchain data layer, each distributed node encapsulates the business activity information received over a period of time into a data block and links it to the longest blockchain in the current blockchain network, forming a new block. At the same time, with the increasing channels for obtaining information on enterprise business activities in blockchain, the amount of data is increasing. This information may originate from different institutions and fields, and there may be certain differences in the representation of information. Its credibility and information value cannot be measured, making it difficult to determine the overall quality of blockchain.

 Include a section that compares your work with previous studies and techniques. This comparison can provide additional support for the validity and novelty of your research. Highlight similarities, differences, and advancements over prior work.

Thanks for the reviewer 's advice. We have added a description in the last paragraph of Section 2 explaining the results achieved by building the new model as follows.

In summary, this paper proposes a multi-source heterogeneous blockchain quality assessment model for enterprise business activities by addressing the shortcomings of existing blockchain enterprise business activity information assessment methods and considering the consistency, credibility and value of data [21] and improve the efficiency of blockchain consistency assessment and considered the credibility of content, while achieving good results in situations with low information uncertainty.

 Suggest a more in-depth analysis of security and privacy concerns associated with multi-source blockchain data. Encourage the inclusion of case studies or practical examples to demonstrate the model's application in real business scenarios and also Propose collaborations with blockchain enterprises for real-world testing and validation of the model.

Thanks for the reviewer 's advice. 

---

## [Decision Letter · Decision Letter 1]

4 Apr 2024

PONE-D-23-32614R1Multi-source heterogeneous blockchain data quality assessment model for enterprise business activitiesPLOS ONE

Dear Dr. Wang,

Thank you for submitting your manuscript to PLOS ONE. After careful consideration, we feel that it has merit but does not fully meet PLOS ONE’s publication criteria as it currently stands. Therefore, we invite you to submit a revised version of the manuscript that addresses the points raised during the review process.

We look forward to receiving your revised manuscript.

Kind regards,

Asadullah Shaikh, Ph.D.

Academic Editor

PLOS ONE

Reviewers' comments:

Reviewer's Responses to Questions

**Comments to the Author**

1. If the authors have adequately addressed your comments raised in a previous round of review and you feel that this manuscript is now acceptable for publication, you may indicate that here to bypass the “Comments to the Author” section, enter your conflict of interest statement in the “Confidential to Editor” section, and submit your "Accept" recommendation.

Reviewer #1: All comments have been addressed

Reviewer #3: (No Response)

Reviewer #4: (No Response)

2. Is the manuscript technically sound, and do the data support the conclusions?

Reviewer #1: Yes

Reviewer #3: Partly

Reviewer #4: Yes

3. Has the statistical analysis been performed appropriately and rigorously? 

Reviewer #1: Yes

Reviewer #3: I Don't Know

Reviewer #4: (No Response)

4. Have the authors made all data underlying the findings in their manuscript fully available?

Reviewer #1: Yes

Reviewer #3: No

Reviewer #4: (No Response)

5. Is the manuscript presented in an intelligible fashion and written in standard English?

Reviewer #1: Yes

Reviewer #3: No

Reviewer #4: Yes

6. Review Comments to the Author

Reviewer #1: Dear

I wanted to inform you that the work you presented is commendable and well-crafted. The previous errors have been addressed effectively, and I find that it has added more clarity and completeness to the research. I look forward to seeing further development in this work.

Thank you very much for your efforts.

Best regards,

Reviewer #3: The following comments are regarding the version that is called Revision 1.

1. Language is poor, enhance the language and check the miss typos. For example, many words are separated into 2 parts, i.e. (s hortcomings) where “s” separated from “hortcomings”, and so on.

2. The introduction needs to be restructured and be more informative to guide the reader, starting with the basic concepts and fundamental definitions and then gradually going through the more related problems and terms. The basic definitions and important clarifications in the introduction should have references. Also, there is redundancy even inside the introduction.

3. While the research tends more toward the enterprise blockchain data layer rather than traditional) blockchain data layers, it is necessary to includes in the introduction an explanation about this distinction ((with references)).

4. Many sentences are long and take place over three or more lines. It is preferable to shorten the length of sentences, for example, by breaking them into 2 or more sentences.

5. The author, in introduction wrote “Large domestic and foreign enterprise units ..”, such sentence is not clear!, at first which of the mentioned example are domestic ? and domestic relating for whom?, and what is the meaning of “units” in such contexts?.

6. In the introduction the author mention that “ … Google, Baidu and Alibaba, have established their own enterprise federated blockchain systems”. But they do not provide reference for such examples! For instance, it is well known that Google's offers Blockchain-as-a-Service (BaaS) which is primarily support public blockchains, but the Authors need to provide references supports their examples about federated blockchains.

7. As a continuation of the previous note, the authors already mention (federated blockchains) without explanation, and why it is related to their work. So, the authors also prefer to insert brief definitions of Public, private, and Federated Blockchains in the introduction, making clear the differences between them.

8. In the related works, the author listed some references under works that is specifically (In blockchain information consistency evaluation), but it not!!! For example, (and not limited to), the reference number (14) is related for natural language processing and the author of (14) dose not mentions any explicitly relationship with Blockchain!!!. Without a clear connection or application area identified within the blockchain, the direct applicability is limited. ((((The authors should reanalyze all the related works again and focus on the similarities within the area of challenges to determine the feasibility of such an application)))). The author should recheck all the related works!. All of them should have direct relationship with the proposed work.

9. I could not find some cited papers in any resource in the internet !!! for example (Disambiguation method of inconsistent records based on factor graph)!!!???? Please provide the correct link, and explain this. I miss something here?!!!!.

10. In the related works, the authors cited the preprint version of a paper (the reference number 12) while this work have official published version: https://aclanthology.org/N18-1071/ . The authors need to focuses on the officially published version and analysis it again.

11. Any fundamental used terms, for example (and not limited to) the triple structure, better to be well defined in the introduction or in the related works with citations

12. The third section needs to reconsider the structure of its sections. The logical connection between its sub-sections must be clearer. Perhaps there is a need for a workflow in the form of an illustration (new figure) or something similar.

13. Let sub-section 3.7 have more informative title.

14. It is necessary to provide an explanation of each data set used in the experiment, along with a reference or link to these data sets.

15. It is important to provide a clear explanation regarding the comparison with reference to the related works, , include citations.

16. In fact, I cannot agree with what is stated in the conclusion, because there is a major weakness in the study of related works.

Reviewer #4: Comment # 1: The second contribution indicated in the introduction should be looked at and rewritten well. it's confusing which basis that statement is based on.

Comment #2 The summary of the introduction should be rewritten because it is too long and makes it difficult to get what you are communicating. some more sentences are long making reading them difficult, and they should be looked at.

Comment # 3: The literature survey must be comprehensive. Add more existing literature

comment# 4: it will be appreciated if the experimental dataset is made available on a publicly accessible repository with a correct or accessible doi. the one provided here can not be assessed.

Comment # 5: In the references section, all references must be of the same style and must be complete eg. Check reference # 21

7. PLOS authors have the option to publish the peer review history of their article (what does this mean?). If published, this will include your full peer review and any attached files.

Reviewer #1: **Yes: **Atheer Alrammahi

Reviewer #3: No

Reviewer #4: No

---

## [Author Response · Author response to Decision Letter 1]

6 May 2024

Responses to Reviewers’ Comments

Dear Editor and Reviewers,

Thanks for your comments concerning our manuscript entitled “multi-source heterogeneous blockchain data quality assessment model for enterprise business activities". The comments are very valuable and helpful for revising and improving our manuscript. We have carefully revised the manuscript and responded all comments below. The changes will not affect the content and framework of the paper. The revised sentences are highlighted in yellow in the manuscript.

Best regards,

Baoyan Song

Reviewer 1:

I wanted to inform you that the work you presented is commendable and well-crafted. The previous errors have been addressed effectively, and I find that it has added more clarity and completeness to the research. I look forward to seeing further development in this work.

Response：

First of all, please allow me to express my sincerest gratitude to you. Thank you for taking the time out of your busy schedule to carefully review and evaluate my paper. 

Once again, thank you for your attention and support for my paper. Your professional spirit and rigorous attitude have left a deep impression on me, and they also inspire me to continue striving for excellence in my future academic pursuits. I hope to have the opportunity to engage in more academic exchanges with you and jointly promote the development of related fields.

Finally, please continue to follow the progress of my paper. If there are any questions or further modifications needed, please feel free to let me know. I will make the necessary adjustments as soon as possible to ensure the quality and accuracy of the paper.

Thank you once again for your careful guidance and valuable opinions! Wishing you a smooth work and good health!

Reviewer 3: 

1. Language is poor, enhance the language and check the miss typos. For example, many words are separated into 2 parts, i.e. (s hortcomings) where “s” separated from “hortcomings”, and so on.

Response：

Thanks for the reviewer 's advice. We have made revisions to the entire text and improved the language expression. The specific modifications are listed below.

Blockchain-based applications are becoming more and more widespread in business operations. In view of the shortcomings of existing enterprise blockchain evaluation methods, this paper proposes a multi-source heterogeneous blockchain data quality evaluation model for enterprise business activities, so as to achieve efficient evaluation of business activity information consistency, credibility and value. This paper proposes a multi-source heterogeneous blockchain data quality assessment method for enterprise business activities, aiming at the problems that most of the data in enterprise business activities come from different data sources, information representation is inconsistent, information ambiguity between the same block chain is serious, and it is difficult to evaluate the consistency, credibility and value of information. The method firstly proposes an entity information representation method based on the Representation learning for fusing entity category information (CEKGRL) model, which introduces the triad structure of related entities in blockchain, then associates them with enterprise business activity categories, and carries out similarity calculation through contextual information to achieve blockchain information consistency assessment. After that, a trustworthiness characterization method is proposed based on information sources, information comments, and information contents, to obtain the trustworthiness assessment of the business. Finally, based on the information trustworthiness characterization, a value assessment method is introduced to assess the total value of business activity information in the blockchain, and a blockchain quality assessment model is constructed. The experimental results show that the proposed model has great advantages over existing methods in assessing inter-block consistency, intra-block activity information trustworthiness and the value of blockchain.

2. The introduction needs to be restructured and be more informative to guide the reader, starting with the basic concepts and fundamental definitions and then gradually going through the more related problems and terms. The basic definitions and important clarifications in the introduction should have references. Also, there is redundancy even inside the introduction.

Response：

Thanks for the reviewer 's advice. We have reorganized and revised the contents of the Introduction to make it specifically better logical and readable.

Blockchain is divided into public chain, private chain, and consortium chain based on its degree of decentralization. Public chain is completely open and transparent, not controlled by any organization, and everyone can participate; The write permission of private chains only belongs to individuals or a certain organization, with a high degree of centralization; The alliance chain, on the other hand, falls between the two and is only open to specific group organizations [8]. Participants can conduct transactions or access information, but only nodes in the alliance have the right to perform transaction verification, publish contracts, and other functions.

3. While the research tends more toward the enterprise blockchain data layer rather than traditional) blockchain data layers, it is necessary to includes in the introduction an explanation about this distinction ((with references)).

Response：

Thanks for the reviewer 's advice. We have added a description in the introduction and add a reference 8. The specific modifications are listed below.

Blockchain is divided into public chain, private chain, and consortium chain based on its degree of decentralization. Public chain is completely open and transparent, not controlled by any organization, and everyone can participate; The write permission of private chains only belongs to individuals or a certain organization, with a high degree of centralization; The alliance chain, on the other hand, falls between the two and is only open to specific group organizations [8]. Participants can conduct transactions or access information, but only nodes in the alliance have the right to perform transaction verification, publish contracts, and other functions.

4. Many sentences are long and take place over three or more lines. It is preferable to shorten the length of sentences, for example, by breaking them into 2 or more sentences.

Response：

Thanks for the reviewer 's advice. We have revised the full text description by breaking down long sentences into short ones, making the paper more readable. The specific modifications are listed below.

Moreover, traditional evaluation methods do not make full use of the features of blockchain. Blockchain features include leaving traces throughout the process, being non-temper able, and traceable. Evaluation efficiency and accuracy are low with traditional methods. This results in the inability of enterprise users and relevant regulatory authorities to quickly screen out suitable blockchains. Establishing a unified analysis model becomes challenging [9].

5. The author, in introduction wrote “Large domestic and foreign enterprise units ..”, such sentence is not clear!, at first which of the mentioned example are domestic ? and domestic relating for whom?, and what is the meaning of “units” in such contexts?.

Response：

Thanks for the reviewer 's advice. We change the description in the introduction. The specific modifications are listed below.

Many IT companies around the world, such as IBM, Baidu and Alibaba, have established their own enterprise federated blockchain systems. 

6. In the introduction the author mention that “ … Google, Baidu and Alibaba, have established their own enterprise federated blockchain systems”. But they do not provide reference for such examples! For instance, it is well known that Google's offers Blockchain-as-a-Service (BaaS) which is primarily support public blockchains, but the Authors need to provide references supports their examples about federated blockchains.

Response：

Thanks for the reviewer 's advice. We have changed the description. The specific modifications are listed below.

Many IT companies around the world, such as IBM, Baidu and Alibaba, have established their own enterprise federated blockchain systems.

7. As a continuation of the previous note, the authors already mention (federated blockchains) without explanation, and why it is related to their work. So, the authors also prefer to insert brief definitions of Public, private, and Federated Blockchains in the introduction, making clear the differences between them.

Response：

Thanks for the reviewer 's advice. We have added a description in the introduction. The specific description is listed below.

Blockchain is divided into public chain, private chain, and consortium chain based on its degree of decentralization. Public chain is completely open and transparent, not controlled by any organization, and everyone can participate; The write permission of private chains only belongs to individuals or a certain organization, with a high degree of centralization; The alliance chain, on the other hand, falls between the two and is only open to specific group organizations [8]. Participants can conduct transactions or access information, but only nodes in the alliance have the right to perform transaction verification, publish contracts, and other functions.

8. In the related works, the author listed some references under works that is specifically (In blockchain information consistency evaluation), but it not!!! For example, (and not limited to), the reference number (14) is related for natural language processing and the author of (14) dose not mentions any explicitly relationship with Blockchain!!!. Without a clear connection or application area identified within the blockchain, the direct applicability is limited. ((((The authors should reanalyze all the related works again and focus on the similarities within the area of challenges to determine the feasibility of such an application)))). The author should recheck all the related works!. All of them should have direct relationship with the proposed work.

Response：

Thanks for the reviewer 's advice. We have reanalyzed all the related works again. The specific modifications are listed below.

This paper introduces a novel approach: a multi-source, heterogeneous blockchain quality assessment model tailored for enterprise business activities. It addresses the limitations of current methods by prioritizing consistency, credibility, and data value [22]. This model enhances blockchain consistency assessment efficiency while also factoring in content credibility, yielding promising results even in scenarios with high information uncertainty.

9. I could not find some cited papers in any resource in the internet !!! for example (Disambiguation method of inconsistent records based on factor graph)!!!???? Please provide the correct link, and explain this. I miss something here?!!!!.

Response：

Thanks for the reviewer 's advice. We have corrected the incorrectly cited papers. The specific modifications are listed below.

[8] Zheng Peilin, Xu Quangqing, ZHENG Zibin, et al. (2021) Meepo: Sharded consortium blockchain. 2021 IEEE 37th International Conference on Data Engineering, Chania, Greece, 1847–1852

 [12] Yi Yang, Ozan Irsoy, and Kazi Shefaet Rahman. (2018) Collective Entity Disambiguation with Structured Gradient Tree Boosting. In Proceedings of the 2018 Conference of the North American Chapter of the Association for Computational Linguistics: Human Language Technologies, Volume 1, New Orleans, Louisiana. Association for Computational Linguistics, 777–786.

[14] Jiang Jing, Wang Kai, Xu Yueqiang, et al. (2022) Optimal Caching Strategy of Operators Based on Consortium Blockchain [J]. Journal of Electronics & Information Technology, 44 (9): 3043-3050.

10. In the related works, the authors cited the preprint version of a paper (the reference number 12) while this work have official published version: https://aclanthology.org/N18-1071/ . The authors need to focuses on the officially published version and analysis it again.

Response：

Thanks for the reviewer 's advice. We have reanalyzed and cited reference 12.

[12] Yi Yang, Ozan Irsoy, and Kazi Shefaet Rahman. (2018) Collective Entity Disambiguation with Structured Gradient Tree Boosting. In Proceedings of the 2018 Conference of the North American Chapter of the Association for Computational Linguistics: Human Language Technologies, Volume 1, New Orleans, Louisiana. Association for Computational Linguistics, 777–786.

11. Any fundamental used terms, for example (and not limited to) the triple structure, better to be well defined in the introduction or in the related works with citations.

Response：

Thanks for the reviewer 's advice. The concept of triple structure is proposed in this article. In order to distinguish from the existing definition, this article is modified to "Triple Evaluation Structure".

The triple evaluation structure is a data model widely used for representing information, consisting of three components: subject, predicate, and object. It is extensively applied in fields such as knowledge graphs and the semantic web. The triple evaluation structure is highly flexible and can be employed to represent various types of information. Different combinations of subjects, predicates, and objects can capture rich semantic relationships, making it suitable for diverse knowledge representation. In summary, the advantages of the triple evaluation structure lie in its flexibility, clear semantic representation, scalability, and effective expression of relationships, making it a key technology in the fields of knowledge representation and the semantic web.

12. The third section needs to reconsider the structure of its sections. The logical connection between its sub-sections must be clearer. Perhaps there is a need for a workflow in the form of an illustration (new figure) or something similar.

Response：

Thanks for the reviewer 's advice. We add a logic flowchart to show our technical roadmap.

Fig 1. The Logic flowchart of Showcase Assessment Model technical roadmap

13. Let sub-section 3.7 have more informative title.

Response：

Thanks for the reviewer 's advice. We change the title as “Case analysis of enterprise alliance chain evaluation”.

14. It is necessary to provide an explanation of each data set used in the experiment, along with a reference or link to these data sets.

Response：

Thanks for the reviewer 's advice. We have uploaded the dataset of this paper to the figshare database and added the generated doi to the content of the experimental setup. The specific modifications are as follows.

The experimental dataset consists of two parts, which are official data searched and crawled on the search engine with the subject term of business activities and real data of enterprises, totaling about 210,000 items, and then simple data cleaning is performed by filling in missing values, smoothing or removing outlier points and other operations to get about 200,000 items of data, and the details of the dataset are listed in Table 2. The datasets are available from https://doi.org/10.6084/m9.figshare.25143503.v1.

15. It is important to provide a clear explanation regarding the comparison with reference to the related works, include citations.

Response：

Thanks for the reviewer 's advice. We have improved the literature and analysis of the relevant work, supplemented the latest technologies and methods, analyzed the advantages and disadvantages, added citations throughout the text, and compared them with the methods proposed in this paper in experiments.

This section compares the running efficiency of each model, the horizontal coordinate represents the entity data set in the blockchain, in units of ten thousand, the vertical coordinate indicates the time required to run each model, and the experimental results are shown in Fig 8.

16. In fact, I cannot agree with what is stated in the conclusion, because there is a major weakness in the study of related works.

Response：

Thanks for the reviewer 's advice. We have improved the literature an

---

## [Decision Letter · Decision Letter 2]

21 May 2024

Multi-source heterogeneous blockchain data quality assessment model for enterprise business activities

PONE-D-23-32614R2

Dear Dr. Wang,

We’re pleased to inform you that your manuscript has been judged scientifically suitable for publication and will be formally accepted for publication once it meets all outstanding technical requirements.

Kind regards,

Asadullah Shaikh, Ph.D.

Academic Editor

PLOS ONE

Additional Editor Comments (optional):

Reviewers' comments:

Reviewer's Responses to Questions

**Comments to the Author**

1. If the authors have adequately addressed your comments raised in a previous round of review and you feel that this manuscript is now acceptable for publication, you may indicate that here to bypass the “Comments to the Author” section, enter your conflict of interest statement in the “Confidential to Editor” section, and submit your "Accept" recommendation.

Reviewer #3: All comments have been addressed

Reviewer #4: All comments have been addressed

2. Is the manuscript technically sound, and do the data support the conclusions?

Reviewer #3: Yes

Reviewer #4: Yes

3. Has the statistical analysis been performed appropriately and rigorously? 

Reviewer #3: Yes

Reviewer #4: Yes

4. Have the authors made all data underlying the findings in their manuscript fully available?

Reviewer #3: Yes

Reviewer #4: Yes

5. Is the manuscript presented in an intelligible fashion and written in standard English?

Reviewer #3: Yes

Reviewer #4: Yes

6. Review Comments to the Author

Reviewer #3: (No Response)

Reviewer #4: This manuscript is a better revision of the previously mentioned comments. It has clarity now and awaiting future improvements in this work

7. PLOS authors have the option to publish the peer review history of their article (what does this mean?). If published, this will include your full peer review and any attached files.

Reviewer #3: No

Reviewer #4: No

---

## [Editor Report · Acceptance letter]

23 May 2024

PONE-D-23-32614R2 

PLOS ONE

Dear Dr. Wang, 

I'm pleased to inform you that your manuscript has been deemed suitable for publication in PLOS ONE. Congratulations! Your manuscript is now being handed over to our production team.

Kind regards, 

on behalf of

Prof. Asadullah Shaikh 

Academic Editor

PLOS ONE